# Erythropoietin: A Personal Alice in Wonderland Trip in the Shadow of the Giants

**DOI:** 10.3390/biom14040408

**Published:** 2024-03-27

**Authors:** Anna Rita Migliaccio

**Affiliations:** Altius Institute for Biomedical Sciences, Seattle, WA 98121, USA; amigliaccio@altius.org

**Keywords:** erythropoietin, anemia, erythrocytosis, kidney failure

## Abstract

The identification of the hormone erythropoietin (EPO), which regulates red blood cell production, and its development into a pharmaceutical-grade product to treat anemia has been not only a herculean task but it has also been the first of its kind. As with all the successes, it had “winners” and “losers”, but its history is mostly told by the winners who, over the years, have published excellent scientific and divulgate summaries on the subject, some of which are cited in this review. In addition, “success” is also due to the superb and dedicated work of numerous “crew” members, who often are under-represented and under-recognized when the story is told and often have several “dark sides” that are not told in the polished context of most reviews, but which raised the need for the development of the current legislation on biotherapeutics. Although I was marginally involved in the clinical development of erythropoietin, I have known on a personal basis most, if not all, the protagonists of the saga and had multiple opportunities to talk with them on the drive that supported their activities. Here, I will summarize the major steps in the development of erythropoietin as the first bioproduct to enter the clinic. Some of the “dark sides” will also be mentioned to emphasize what a beautiful achievement of humankind this process has been and how the various unforeseen challenges that emerged were progressively addressed in the interest of science and of the patient’s wellbeing.

## 1. Introduction

It is very difficult for a student who starts their medical education today to understand what it was like to study blood in the last century. The clinical interest in studying blood was extremely high because blood loss had been identified as a major cause of death since the very beginning of medicine. Among the first physicians who reportedly understood the relationship between major interruptions of blood flow and death was Alcmaeon, a physician trained in the medical school of the Greek colony of Croton in South Italy in the 6th century BC. In addition, purging the body from “poisoning humors” by blood-letting was a clinical practice that originated in ancient Egypt, already discussed by Hippocrates and implemented by Galen in the ancient Roman Empire. In these ancient Roman times, the relationship between blood and death Donewas so well established that lethal physician-initiated blood-letting was a common suicidal practice often performed upon the Emperor’s order. Although debated, the use of blood-letting remained a popular clinical remedy to treat a wide variety of illnesses until the beginning of the last century. By contrast, attempts to prevent excessive blood loss can be traced back to the early 1900s [see [1]], but the first successful husband-to-wife blood transfusion to treat postpartum hemorrhage was performed by Dr. James Blundell in 1818 [2], and transfusion became a widely used medical practice only after the discovery of the A, B, and O human blood types by Karl Landsteiner in 1901 [3]. For the importance of his discovery, Dr. Landsteiner received the Nobel Prize in Physiology or Medicine in 1930.

That blood contained red blood cells was discovered in the early 17th century by the microscopic observations made by Antoni van Leeuwenhoek, a Dutch cloth merchant with a passion for optics. However, the mechanism(s) that regulates their production in vivo and attempts to exploit them to increase their insufficient numbers (i.e., to treat “anemia”) remained a mystery for some time. Anecdotical treatments of anemia by “stimulators” had already been made by the physicians of the Magna Grecia in South Italy, who used “iron sault” scraped from reddish stones. The debate whether, in addition to transfusion, anemia could be treated by promoting red blood cell production by the body with some type of “substance” began to gather momentum at the beginning of the last century. The hypothesis that the plasma contained a “factor”, defined as hemopoietin, responsible for stimulating the production of new red blood cells was put forward in 1905 by Dr. Paul Carnot and his graduate student, Clotilde Deflandre, of the University in Paris [4]. The routine experience that altitude, more precisely the levels of O_2_ in the air, reversibly modulate only the number of the red blood cells present in the blood led Drs. Eva Bonsdorff and Eeva Jalavisto in 1948 to define more precisely this factor as ‘erythropoietin’ (EPO) [5]. The formal definition that the plasma from anemic rabbits would increase reticulocyte counts in normal rabbits in a concentration-dependent manner was then provided by Dr. Allan Erslev in 1953 [6], and finally Dr. Jacobson et al. reported in 1957 that EPO is mostly produced by the kidney, the organ in the body best suited to measure the need of the organism for new red blood cell production because it extracts the least oxygen with nearly the greatest blood flow, so that the EPO-producing organ itself is not subjected to metabolic demands and skew erythropoiesis [7] (Figure 1). 

This scholarly information is summarized here only to put the field in the context. Additional historical perspective of the development of the concept of EPO is described in greater detail in several excellent reviews, such as that written by Dr. John Adamson (John), who pioneered the first EPO trial, for the book dedicated to the major clinical achievements in Hematology published in 2008 by the American Society of Hematology to celebrate the 50th anniversary of its foundation [8]. 

Bringing a new drug to the clinic not only involves the discovery of the “active principle” but also has two additional facets: the identification of the patient population most suited for “first-in-man” use and the massive, safe, and reproducible production of the drug. In addition, since EPO was the second product produced by recombinant technology to reach the clinic (the first one was insulin), the field had to surmount numerous legal issues across continents. Furthermore, anemia is so wildly diffused (that in modern clinical terminology means “the size of the target population is so large”) and is so debilitating for the patient (that in modern terminology means “it has such great socio-economic impacts), and the blood necessary for transfusion is so scarce and potentially risky (that in modern terminology means anemia is an unmet clinical need), that an effective curative “drug” was predicted to be extremely profitable. Large “profits” come with “dark sides” and with “losers” and “winners”. Below is a summary of how the various facets of the EPO saga evolved, including its “dark sides” and the “winners” and “losers”, for which I have direct or indirect personal knowledge.

## 2. The Discovery of the Active Principle

Once the proof-of-principle that plasma contains a “soluble factor” which stimulates erythroid production was established by Dr. Allan J. Erslev [6], moving the field forward required establishing a biological assay to identify its biochemical nature and purification.

The first quantitative biological assay used to test EPO was developed by Dr. Erslev and was used mostly as a differential diagnostic tool to discriminate the cause underlying different anemic and polycythemic conditions (reviewed [9]). For example, it was used to assess whether polycythemia was associated or not with high EPO levels, i.e., whether it was due to forms, possibly inherited, of dysregulated EPO production or to cell-autonomous activation of erythroid differentiation by somatic driver mutations such as those found in the myeloproliferative neoplasm Polycythemia Vera [10]. This bioassay was represented by a mouse model of blunt EPO production, was long, cumbersome, expensive, and had low sensitivity. Its rationale was provided by the observation that in mammals, exposure to the low O_2_ pressure of a hypobaric chamber stimulates an adaptation mechanism that includes greater production of EPO by the kidney to increase the hematocrit to the levels necessary to normalize the delivery of the gas to the tissues. When the animals are brought back to normoxic conditions, they blunt the production of their endogenous EPO until their hematocrit returns to normal levels. Therefore, they produce new reticulocytes only when exposed to exogenous EPO. Using this cumbersome bioassay, Dr. Eugene (Gene) Goldwasser succeeded in the herculean task of demonstrating that EPO was a protein and, after numerous attempts that lasted as many as 17 years, purified it to homogeneity [11].

The eventual purification of EPO to homogeneity was greatly favored by the development of more user-friendly, less expensive, and more sensitive bioassays. The bioassay that quickly replaced in vivo determinations in mice exposed to hypoxia was an in vitro titration curve of the number of colonies developed by the erythroid colony-forming cells (CFU-E) present in the mouse bone marrow. This assay is based on the observation made by Dr. Connie Eaves, known at the time as Dr. Connie Gregory [12], and confirmed by Dr. Norman Iscove [13] that CFU-E are erythroid progenitors that grow in vitro only in the presence of EPO. This bioassay defines one arbitrary unit of EPO as the concentration that sustains the growth of a plateau number of colonies from a given number of mouse bone marrow cells. This in vitro bioassay has many advantages over the in vivo one previously used: it is more sensitive (it requires low amounts of tester material, few uL vs. several mL), is quicker (2 days against 10–12 days), is less expensive (three mice against tens of mice), is more sensitive, and does not require a hypobaric chamber, specialized devices available only in few centers. It has, however, its caveats: it is very sensitive to changes in the density of the semisolid media used to grow the colonies, to the presence of endotoxin, a common bacterial-derived contaminant of products obtained from urine, and to changes in humidity and O_2_/CO_2_ pressure in the atmosphere of the culture incubator. At the time in which I personally made EPO determinations using this in vitro bioassay, these caveats were addressed by testing the performance of all the new culture reagents against that of validated material and running a new control CFU-E assay every single day to ensure that their number would remain constant, indicating that the culture environment had not changed over time. In other words, I had devised an “empirical” and “ante litteram” quality assurance procedure to ensure that determinations of EPO concentrations would be comparable over time. This hard and rigorous work paid back in a way that could not have been foreseen at that time. In fact, many years later, these procedures were incorporated into the quality assurance protocol devised to define the quality of the cord blood units stored in the first cord blood bank for allogeneic transplantation established at the New York Blood Center [14]. Furthermore, they inspired Dr. Allen Eaves to establish Stem Cell Technology, a company dedicated to the formulation of user-friendly culture kits to determine the number of hematopoietic progenitors in hematopoietic tissues. Since the EPO used to standardize the growth of the CFU-E was not pure, the concentration of EPO determined in the test material with this in vitro bioassay was expressed as arbitrary units. 

Finally, when pure EPO and antibodies against EPO became available, it was possible to develop rapid (few hours), unbiased (EPO levels are defined against titration curves based on actual ng amounts of protein and not on its biological activity), and sensitive immunoassays for the protein. The first EPO radio-immunoassays were independently developed in 1979 by the Eugene Goldwasser [15] and Joseph Garcia [16] laboratories using antiserum derived from rabbits immunized with crude human urinary EPO [15]. However, it was not until recombinant EPO became available and antibodies against EPO were purified that clinical-grade radio-immunoassays for EPO became available [17]. Although similar in principle, radio-immunoassays have been currently replaced by ELISA, which uses fluorescently labeled antibodies instead of radioactive probes as tracers. 

## 3. The Long Process to Obtain “Pure EPO”

One of the challenges to be addressed to purify any “biological factor” is to identify a suitable source to be used as starting material. The customary use of transhumance, i.e., the migration from sea level to high altitude, in the upbringing of sheep, made these large animals the ideal source to purify EPO. Experiments in sheep had played a pivotal role in the identification of the kidney as the organ responsible for producing the “factor” that increases the hematocrit under the atmosphere found at high altitudes. The design of these experiments involved observing that the hematocrit would increase more slowly when the animals were brought to high altitude after one of their kidneys had been removed. However, it would rise quickly if the animals were transfused with plasma from another animal of the herd, which had its kidneys intact. It may be suspected that the fact sheep kidneys are delicious and are a regular component of the British breakfast may have also guided the choice to use sheep as the experimental model to purify EPO. A Medline search using “sheep” and “EPO” as keywords retrieved more than 44 papers that were published between 1955 and 1965 on this subject, most of which, unfortunately, are not available online. This wealth of information influenced Dr. Eugene Goldwasser to investigate plasma from transhumance sheep as the source for his first EPO purification attempts. A three-step method involving ion-exchange chromatography, ammonium sulfate precipitation, and batch adsorption that achieved a fraction with an activity of 450 EPO units/mg protein representing a purification factor of about 64,000 over the starting material was published in 1962 [18] and was refined into a method that allowed to purify the protein to homogeneity in 1971 [19]. This purification process was exploited by the Canadian Connaught Medical Research Laboratories to produce the first EPO for commercial sale. This legendary product was named Step III EPO because it purified up to the third step of the process described by Gene, and its availability greatly contributed to establishing in vitro surrogate models to study terminal erythropoiesis in the 1970s. 

The discovery that some of the diseases associated with anemia also increase the amount of EPO present in the blood, which is eventually secreted in the urine, paved the way for efforts, often supported by national funding agencies, to collect urine from anemic patients to be used as a source to purify the human growth factor. The first of these attempts was an effort involving daily collection of urine from 49 patients with anemia secondary to hookworm infection until their hemoglobin levels rose to 6 g/100 mL made in Argentina at the Facultad de Medicina, Universidad Nacional del Nordeste, Corrientes. This effort was coordinated by scientific personnel from the Donner Laboratory of Medical Physics, University of California, Berkeley, CA, USA, and was funded in part by the Consejo Nacional de Investigaciones Cientificas y Tecnicas of Argentina (grant 1991), by the Comite de Investigaciones Medicas de Corrientes, Argentina, and by the United States Atomic Energy Commission [20].

This first attempt was followed by an initiative to create a center to collect urine from patients with aplastic anemia from hospitals all over the US that was active from 1964 to 1980 and was sponsored by the National Heart Lung Blood Institute (NHLBI). This center was directed by Dr. Peter Dukes, who had collaborated with Gene when he was at the University of Chicago from 1954 to 1967 [21]. The urine concentrates prepared by this Center were used by Gene as starting material to purify human EPO. Over the years, Gene regularly published methods to purify human EPO with increasing purity until Christmas morning in 1977; he received a generous gift hand-delivered by Dr. Takaji Miyake of the Kumamoto University, Fukuoka, Japan, of protein concentrates from more than 2500 L of urine from patients with aplastic anemia. This gift allowed him to finally succeed in purifying 8 mg of human EPO [11] and to publish its amino acid sequence in 1986 [22]. The dedicated effort to purify EPO by Gene was continuously supported from 1985 to 1996 by the National Institute of Health (NIH) with the prestigious Research Grants MERIT Award (R37, 5R37HL021676-10). 

Progress made by Gene in the methods to purify human EPO was rapidly implemented by Dr. Peter Duke to produce partially purified material, which was then freely supplied by the Blood Resources Branch of NHLBI to investigators on a project meritorious basis. This progress may be followed in the Materials and Methods sections of the papers on erythropoiesis published at that time, which ranged from 70 IU/mg of protein in papers published in 1976 (see as an example [23]) to 1140 IU/mg of protein in 1986 (see for example [24]). 

When pure EPO became available, Gene began to provide investigators worldwide, on a meritorious basis, with sealed capillary tubes containing 100 IU of pure material each for their research. I still remember the excitement when our laboratory received one of them to perform the first EPO concentration/response curve of human erythroid progenitors present in the human fetal liver [25]. This tube was carefully cut with a scissor, and the material was retrieved with 100 μL of culture medium in a 21G needle-armed insulin syringe. Although pure recombinant EPO became available in 1977, due to its limited amount, it was not until 1998 that its NMR structure in the native and receptor-engaged form was determined [26].

## 4. The European Effort to Purify EPO Supported by the Volkswagen Foundation

Several initiatives to purify EPO were also made in Europe. To facilitate the purification of EPO, the Italian Army, among others, delivered to the USA large amounts of urine concentrates collected from its soldiers. This material, however, was useless because it contained levels of EPO not greater than those present in urine at baseline and was heavily contaminated with endotoxin. 

Among the European initiatives, an attempt worthy of mentioning in detail is an international collaboration supported by the Volkswagen Foundation that was based on an extremely anemic patient from Turkey admitted in 1978 to the Department of Clinical Medicine at the University of Naples, Italy, who was diagnosed with pure red cell aplasia by Dr. Cesare Peschle. Dr. Peschle had just returned from the USA, where he had been trained on the in vivo EPO bioassay by Dr. Albert Gordon and had established a rigorous EPO bioassay method in his laboratory in Naples [27]. He soon recognized that the patient’s urine contained a high level of EPO and started a massive collection program under conditions that we would define today as good laboratory practice. The urine was collected (the patient was asked to urinate in the cold room), concentrated against glycopolyethylene, and lyophilized (specific activity, 20–27 IU/mg of protein) in the cold room. So as not to cause a decrease in the amount of EPO present in the urine, the patient was left untreated for an entire month until he produced enough urine and returned to Turkey. This medical conduct, which would raise many eyebrows today, was allowed by the fact that, at the time, there was no legislation for the protection of patient rights. We believe this case is not unique in the history of bio-recombinant products. It was preceded by an even more egregious failure to recognize patients’ rights, famous because of the outcome of the work at Johns Hopkins University, which led to the development of the HeLa cell line, used by countless investigators worldwide since its generation in the early 1950s [28]. Their existence raised awareness of the importance of developing the current legislation dedicated to the protection of human subject rights during clinical investigations. 

I have first-hand knowledge of this urine collection because my companion in life and research, Dr. Giovanni Migliaccio, spent our honeymoon in the cold room with this patient to ensure that the urine samples were carefully collected under conditions that prevented bacterial growth, a common challenge of protein purification from urine, to ensure that they would be free of endotoxin, a bacterial bioproduct the presence of which, by interfering with the erythroid differentiation process, would have made the material unsuitable for clinical and experimental research. The lyophilized urine proteins were then provided to Dr. Norman Iscove, who succeeded in purifying EPO by means of a three-step chromatography method very similar to that devised by Gene and which produced a product with a specific activity of 50–300 IU/mg of protein (see the acknowledgments [25]). Dr. Iscove published his purification method in 1977 [29]. However, the EPO purified from this single patient did not move the field too far, and it has another “dark side” worth mentioning. Confusion about the bioassay to be used for determining the EPO concentrations in biological samples allowed cheating. The grant from the Volkswagen Foundation was based on the premise that Dr. Peschle would deliver two-thirds of the EPO obtained after purification for use in their clinical testing to the partners. Since EPO titrations based on CFU-E assays are three times more sensitive than those based on mice exposed to hypobaric conditions [30], Dr. Peschle determined the EPO titers of the starting material with the hypoxic mice assay and those of the purified material by the CFU-e assay, allowing him to deliver only one-third of the EPO amount he was supposed to provide according to the contract. Besides suggesting potential professional misconduct practices not uncommon in the EPO saga, this story highlights how difficult it was in the 1970s and 1980s to compare the EPO levels present in the plasma of patients detected across different laboratories, hampering the precision of the clinical diagnosis. 

## 5. Cloning and Production of Recombinant Human Erythropoietin

In the 1980s, progress in recombinant DNA technology and the approval of the recombinant insulin developed by Genentech for clinical use prompted the birth of several small Biotech companies in search of fame and profit by generating additional recombinant factors for therapy. Although most of these Biotechs had “interferons” as their main target, at least nine of them, including Biogen and Genetics Institute in the USA and Kirin in Japan, were interested in the pharmaceutical development of recombinant EPO. The publication of the paper on the purification of human EPO to homogeneity made Gene a very popular “man” among these companies, which competed with each other to bring him in as a consultant. The winner of Gene’s trust was a company named Applied Molecular Genetics (Amgen), located in Thousand Oaks, CA, USA. In 1981, the company entrusted the project of cloning EPO to Dr. Fu-Kuen Lin, a genetic engineer with a previous interest in herbal medicine who had immigrated to the US from Taiwan, and Dr. Joan Egrie, a young, energetic biochemist with expertise in erythropoiesis. Dr. Egrie was also in charge of establishing and coordinating all external collaborators with relevant scientific expertise. The project lagged for two years due to the paucity of pure EPO made available to Dr. Lin by Gene. When finally confronted with the option that Amgen would close the project, Gene delivered material sufficient for protein sequencing. Sequencing was performed by Dr. Lee Hood, then at Caltech, who eventually moved to the University of Washington. The protein sequence was then used to design small, degenerated coding DNA fragments that were used to isolate the genomic DNA from a fetal liver library created by Dr. Maniatis. Dr. Lin then expressed the EPO gene in the Chinese Hamster Ovary cell (CHO) cell line that had already been established as safe by the Food and Drug Administration (FDA) during the process of approving recombinant insulin for clinical use [31]. The availability of the EPO-expressing CHO line paved the way for its mass production under clinical-grade conditions and allowed the first clinical trial to begin. The process of developing recombinant EPO into a clinical product has been summarized by Dr. Joan Egrie in [32]. The business vision that guided the foundation of Amgen, the entrepreneurial effort that sustained its initial operations and prevented its bankruptcy until the commercialization of EPO took place, making Amgen one of the most profitable biotech companies in the world, are summarized in a book by Dr. Gordon Binder, a former Amgen Chief Executive Officer [33]. 

## 6. The First-in-Man Clinical Trial to Treat Anemia of Chronic Kidney Failure

The choice of the target patient population for the first-in-man use of recombinant EPO was brilliant and demonstrated a profound knowledge of the mechanisms that control erythropoiesis by Dr. John Adamson, Chief of Hematology at the University of Washington. The extensive knowledge obtained in animal models and in humans that the kidney is the organ responsible for producing EPO led to the realization that patients with renal failure undergoing chronic dialyses are anemic because their kidney is no longer capable of producing the hormone [34]. Based on this information, Dr. Joseph Eschbach, a nephrologist with a large patient practice in Seattle, and John conceived the hypothesis that “kidney failure” represents the ideal disease model to test recombinant EPO as “hormone replacing therapy” [35]. In other words, to test the hypothesis that providing what the body is no longer capable of producing is effective in eliminating the symptoms of a disease. The proof-of-principle for this hypothesis had already been obtained in 1984 by experiments demonstrating that EPO-rich plasma cures anemia in a sheep model of chronic renal failure [36]. The first clinical trial was approved by the NIH and performed in 1986. It was a dose-escalation study involving 25 patients, 18 of whom received effective EPO doses, and 7 received a placebo. Twelve of the 18 patients who received EPO were transfusion-dependent and became transfusion-independent once they received EPO. This study followed the treated patients for several months and was finally published as an academic–biopharma collaboration in January 1987 [37]. 

When I discussed with John why he thinks Amgen had chosen the University of Washington over the several clinical centers that had volunteered to conduct the trial, he said, “I think it was because we had such a history in iron metabolism/kinetics (thanks to the studies performed by Dr. Clement Flinch, the clinician who built the Hematology division in the University of Washington) [38] which our competitors lacked. And iron studies were among the keys to giving/monitoring EPO’s effect in vivo”. Personally, I believe that Dr. Joeseph Eschbach and John were entrusted with the first trial because they had the strongest “pre-clinical” experience, Joe was a damn good nephrologist, and John had a reputation for being extremely “rigorous”. Although their research on sheep may be considered “boring” when compared to the exciting molecular biology experiments that were being conducted in those years (these are the years in which, as an example, the first oncogenes were discovered), it built the experience necessary for the human trial and their “integrity” reputation necessary to assure that the “results” were believable. John, of course, disagrees that the experiments in sheep are “boring” and adds, “Actually, one of the cool things about the model is the ‘switch’ from hemoglobin (Hbg) A to C, which had been shown to be EPO-dependent [39], which is why we focused the studies on sheep identified as having Hbg A”. 

When asked what he remembers of the first EPO trial, John said, “I do remember one thing about the first clinical trial. For whatever reason, we started with EPO doses that were very low—perhaps because it was all so new. But the first few doses had no effect and the company became worried. Then, the first signal—and the rest is history”. By contrast with this understated recollection, Dr. Rebecca Haley, the medical fellow who was attending the patients, has a more exciting recall of this first trial. She said, “once the right EPO dose was reached, the study was no more double blind because there could be no mistake on the patients who were receiving EPO”. One last story to put things into perspective recalled by John is that Dr. George Rathmann, who headed Amgen at the time of the EPO trial, was diagnosed with myeloma later in life and eventually developed renal failure and was the beneficiary of his own company’s product [40].

The NIH provided continuous and effective support at all stages of this trial and assisted in its design, making sure that the results would meet the criteria to support the approval of the drug for clinical use. In fact, against John’s advice, the NIH mandated transfusion independence, and not hematocrit levels, as the primary clinical endpoint of the study. This was because it knew that down the road, to be able to approve the clinical use of EPO, the FDA would require assurance that the “product” was an effective alternative to blood transfusion, a therapy that was becoming increasingly challenging to provide due to product limitation and safety concerns. In addition, the NIH provided consistent economic support, as demonstrated by the acknowledgment section of the paper, which recognizes not only the excellent technical assistance of Mrs. Glenda Schneider and the financial support by Amgen but also research (AM-19410 and AM-33488) and Clinical Research Center (FR-0037) grants from the National Institutes of Health [37].

Another interesting twist of the saga is the reason why it was so easy to recruit patients for this first trial with a recombinant growth factor in Washington State. In the 1980s, the discovery of blood-borne pathogens, one of which was the AIDS virus, raised great concerns about the safety of blood transfusions. These concerns were especially high in the State of Washington, where the first clinical trial with EPO was conducted. In fact, to favor the development of bone marrow transplantation by the Fred Hutchinson Cancer Center, the State of Washington had restricted the practice of dedicated blood donation for transfusion, i.e., the transfusion to a patient of blood donated by dedicated volunteers. In addition, since patients recovering from bone marrow transplants are immune-depressed and sensitive to infections, blood banks in this state have implemented the practice of reserving the “safest blood” for these patients. As a result, less fragile populations in need of regular blood transfusions, such as patients with chronic kidney failure, had a greater incidence of post-transfusion-related infections, including hepatitis B, in Washington State than those in other US states. The chronically transfused patients were so aware of this risk that Dr. Eschbach had no problem recruiting large numbers of patients with kidney failure in Seattle for the EPO studies. 

## 7. EPO Becomes a Worldwide Pharmaceutical Business

As soon as the preliminary data became available, Amgen was confronted with two problems: (1) How to produce a large amount of clinical-grade product foreseen to be necessary to satisfy the vast patient population likely to benefit from the treatment, and (2) how to secure the ability to make it available worldwide. 

The second problem was easy to solve. Amgen established a partnership with a group in the United Kingdom (UK) that, using the information on the Eschbach/Adamson trial while it was still unpublished, designed and performed a single dose trial in a small number (10 patients, only 4 of whom were transfusion-dependent) of anemic patients with kidney failure in the UK. The follow-up of this European study was very short, and it allowed the investigators to quickly publish their results in the Lancet in November 1986 [41]. Although clinically less strong than that performed in the US, this study provided the green light to start the procedures necessary to obtain authorization from the European Medicines Agency (EMEA) to use EPO for the treatment of anemia of patients with kidney failure in Europe. 

A side note of this story is that although the clinical trial with EPO on transfusion-dependent patients in the US had started much earlier than that in Europe, the European trial was published two months and one year earlier than the US trial (November 1986 vs. January 1997). The publication of the results of the US trial had been delayed because most scientific journals at that time did not have an authors’ conflict of interest policy to reduce the perception that the results of the study had been biased by the possible conflict of interest of the authors, who were Amgen employees. To address this caveat, Scientific Journals and Scientific Congresses worldwide have implemented the rule that all papers/communications should include full disclosure of possible conflicts of interest by the authors. 

The first problem was more complicated to solve because it required establishing partnerships with suitable pharmaceutical partners. To cover the EPO needs of the US and European patients, Amgen, whose trade product name was Epogen, established a corporation with Ortho Pharmaceutical Corporation (Raritan, Franklin Township, NJ, USA), which became responsible for mass-scale production of the product. To ensure that EPO manufactured by Ortho, which was named Procrit, was as effective as Epogen, which had been used in the first clinical trial, Dr. Eschbach and John performed a dedicated clinical study [42]. The interest in the results of this trial was so relevant for the patients that NIH was willing to increase its financial support for the study. As further proof of his integrity, the only request by John was to add to his grant a fellowship to cover the salary of Dr. Rebecca Haley, the hematology fellow dedicated to the patients’ care. Most of the EPO that was instead sold in Europe was produced by Roche under the name NeoRecormon. Later on, Amgen developed a second-generation recombinant EPO, commercialized by the name of Aranesp, created in the laboratory to allow re-patenting by adding carbohydrate side chains to increase the half-life and thereby reduce the frequency of the administration. 

The market in East Asia was instead assured by a partnership established by Amgen with Kirin Pharmaceutical, the biotech branch of the giant Japanese Kirin Brewery Company, Tokyo, Japan. The development of the Amgen/Kirin Pharmaceutics partnership is summarized in a chapter by Mrs. Michael Lyndskey, who had interviewed many of the protagonists, including Dr. Akihiro (Sam) Shimosaka, the Vice President of the Licensing Department of the Pharmaceutical Division of Kirin Brewery Company at that time [43]. 

The Kirin Brewery Company is a large, long-established, and successful firm in a traditional old economy sector in Japan. Its origins date back to 1870, and it is the oldest Japanese brewing company. It changed owners and names several times, and finally, foreign investors entrusted engineers from the UK and experienced brewers from Germany to renovate the production plants and devise new brews, respectively. In 1888, the company launched its most successful beer named “Kirin”, after the mythological Japanese dragoon that brings good fortune. Financial support and management from Mitsubishi Zaibatsu led them to formally establish the Kirin Brewery Company in 1907, which soon started to export its product in the US and other states outside Japan. 

Although its core business was based on the traditional biotechnology of fermentation, in 1980, the company’s policy changed dramatically so as to diversify into modern biotechnology and enter into the “new economy” business of biopharmaceuticals funded by Kirin Pharmaceutical. From its beginning, the new company had included EPO as one of the most relevant biopharmaceutical targets of its portfolio, and Dr. Fumimaro Takaku, a world-renowned hematologist at the University of Tokyo, who had played a pivotal role in the development of Hematology in Japan [44], among its consultants (Figure 2). 

Dr. Takaku had been exposed to research on EPO during a sabbatical in Gene’s laboratory at the University of Chicago [46]. His laboratory then became a hub where researchers from Kirin learned erythropoiesis and the biochemistry of EPO [47,48]. Despite the strong effort of the Kirin researchers, the cloning of EPO was lagging until the opportunity of “international scientific collaboration” provided by the International Society of Experimental Hematology (ISEH) came to the rescue. ISEH was established in 1950 by hematologists around the world to accelerate the search for remedies to address the enormous clinical needs raised by the aftermath of the atomic bombings of Hiroshima and Nagasaki. All the scientific players of the EPO saga were prominent figures of this society; for example, Dr. Takaku co-edited a book dedicated to advances in bone marrow transplantation sponsored by the Society [49]; Dr. Dukes became treasurer of ISEH in 1972 and covered this position for over 20 years, and John was a devoted member of the Society and was elected as President in 1993. These investigators had the opportunity to discuss the various facets of EPO during the annual meetings of the Society. Kirin became aware that Amgen had cloned EPO when the news was published in 1984 in the January issue of the official journal of the society, Experimental Hematology [43]. In spite of its small size (in 1981–1982, Amgen Staff consisted of 42,100 people), Kirin was eager to enter into a commercial partnership with this Biotech. Dr. Shimosaka made the first phone calls in 1983 and made the first visit to Amgen in the USA in February 1984. During this meeting, he succeeded in convincing the leadership of Amgen that, in addition to research and development, the driving of a pharmaceutical company requires marketing expertise. Dr. Shimosaka was very persuasive, and this first meeting was followed by a second one held in Japan that led quickly (in three days) to the signature of a joint venture contract stating that Amgen would cover the costs of the production of EPO in the USA and give Kirin the rights to market EPO for patients with chronic kidney failure in Japan, while Kirin would compensate Amgen for the knowhow on recombinant EPO and cover all the development costs in Japan. On a money basis, the agreement involved an investment of USD 24 million dollars divided 50:50 between the two parties: 12 million dollars from Kirin and 4 million dollars, plus the proprietary know-how required to manufacture EPO valued at 8 million dollars from Amgen [43]. In addition, Kirin would provide Amgen roller-bottling and manufacturing scale-up technology developed for its brewing business, which is necessary for mass production of clinical-grade EPO. Kirin also managed all the pre-clinical and clinical studies necessary to obtain the authorization for the clinical use of EPO in Japan [50].

Recombinant EPO, including the manufacturing process developed by Kirin based on roller-bottled cell culture, was approved by the FDA in 1989 (Figure 1). As the product reached the market, the initial estimates of its worldwide market of USD 100 million dollars per year were soon surpassed as the EPO sales reached USD 97 million dollars already in 1989 (only three years after the publication of the first trial), USD 276 million dollars in 1990, and USD 409 million dollars in 1991. Soon, EPO became the world’s fastest-selling drug in terms of annual sales and, in 2002, exceeded USD 7.8 billion dollars [43]. 

Fifty years have passed since the cloning of the EPO gene. It is possible then to assess who the big “losers” and “winners” of the saga were. The companies (in primis Amgen and Kirin and the additional pharmaceutical firms that contributed to its mass production for the rapidly expanding clinical market) and their Chief Executive Officers (CEO) were undoubtedly big economic “winners”. The story also ends well for the Amgen personnel who worked on the scientific site of EPO development. I knew Drs. Egrie and Lin on a personal basis as we meet regularly at social events during the annual meetings organized by ASH and ISEH. Dr. Lin, who also cloned the monkey EPO gene to facilitate in vivo experiments in baboons [51], was very kind to me and wrote my name in Chinese, choosing for it the ideogram which means “ivory tower”. Dr. Egrie was always very interested and supportive of my research. She invited me to give a few seminars at Amgen and provided me with recombinant EPO and other growth factors that lasted for several years (see, as an example, [52]). In addition, with salary support, Amgen compensated Dr. Lin and Egrie with stocks in the company so that their eventual premature retirement saw them financially secure and free to enjoy their life. The story ends less well for the scientists whose primary goal was recognition by their peers. This recognition usually ended with their obituary published in prestigious journals [53,54,55]. For the few who defeated age and are over 80 but alive, their peers organize regular celebrative retreats. As an example, past fellows organized for John a dinner at ASH 2014 to celebrate the mentor award that had been conferred to him by the society a year earlier. Many of the hosts of this dinner have had a primary role in the EPO saga (Figure 3). 

More recently, in 2023, the surviving protagonists organized a retreat at ASH to celebrate the achievements of EPO (email to ARM by Dr. Shimosaka on 17 December 2023). From this summary, it appears that scientists were not, in general, the big “winners”. However, I am sure that eventually, time will put things into perspective again and that the role of these investigators will be accurately acknowledged in Hematology textbooks. And then there are the “collaborators”, the loyal and skilled crews that, like Drs. Miyake, Powell, Haley, and myself, made things happen in the shadow of the giants. These crews worked hard, and their contribution was recognized in the publications, but they often acted as flower girls and boys in the big game. Some of them, like Drs. Miyake [61] and Powell [57,62], tried to become independent and have a protagonist role with no success. These people may be considered the big “losers”. Others, like Dr. Haley and myself, kept low profiles and are content that the end of their career was accompanied by the consciousness that their work had an impact that will last for many years to come. 

## 8. The Year of the Successful Clinical Trial with EPO the Nobel Prize for Biopharmaceutics Was Awarded for Research on Nerve Growth Factor

In January 1986, a few months before the publication of the first EPO clinical trial, the Nobel Prize in Physiology or Medicine was awarded to Drs. Stanley Cohen and Rita Levi Montalcini for their discovery of nerve growth factor (NGF). It is an apparent paradox that the year that the first recombinant growth factor made its clinical debut, the Nobel Prize was awarded for the discovery of NGF, which, until very recently, has had limited clinical applications [63]. It is possible that the involvement of the pharmaceutical industry (Amgen) and the dispute over the patent rights for cloning EPO (see below) may have contributed to reducing the enthusiasm of the Noble Committee toward EPO. 

As mentioned in Dr. Don Wojchowski’s obituary [55], Gene was a sweet person and a generous scientist who was always ready to engage with young students interested in erythropoiesis and who made an effort to regularly follow their subsequent scientific progress. He also demonstrated a great interest in my own research, and years later, in one of our regular conversations over lunch, I plucked up the courage to ask him about his scientific interaction with Rita Levi Montalcini and how he felt about NGF and not EPO being recognized with the Nobel award. He revealed with a smile that over the years, Rita and he had been reporting the progress made on the purification of NGF and EPO, respectively, to the same scientific meetings. He volunteered to say that he personally liked Rita and he had always enjoyed their regular scientific interaction. It had been a continuous source of frustration for him that, due to an easier source (the saliva produced by a transplantable murine salivary gland vs. urine) and a more manageable bioassay (nerve outgrowth in chick embryos vs. the in vivo ex-hypoxic mouse assay), NGF was purified to homogeneity (NGF in 1954 and EPO in 1977) and cloned (NGF in 1983 and EPO in 1985) several years before EPO [64,65,66]. Due to these timing considerations, he felt good and completely understood why Rita, and not him, had won the Nobel prize. What he was somehow still resenting was the fact that, thanks to his information, Amgen had probably already cloned EPO in 1982 but decided to concentrate on moving forward with the clinical trial and delayed the publication of the cloning until November 1985 [31], eight months after the cloning of the gene had already been published by the Genetics Institute [61]. In addition, Gene confided to John, who was one of his dearest friends, that “the day in which he received the Nature magazine and saw the paper on the cloning of EPO was one of darkest day in his life, second only to the day his wife had died, also because the last author of the paper was Dr. Miyake, his former collaborator who had provided him with the urine for his first purification”. Dr. Goldwasser clearly recognized that EPO and his personal life were profoundly intertwined, and more on his lifelong relationship with the hormone may be found in [67]. 

EPO, due to its widespread application to treat anemia, is undoubtedly the recombinant product that has had the greatest clinical impact and generated by far the largest profit worldwide. This fact was finally indirectly recognized by the Nobel Committee in 2019 when the Nobel Prize in Physiology or Medicine was awarded jointly to Drs. William Kaelin, Peter Ratcliffe, and Gregg Semenza “for their discoveries of how cells sense and adapt to oxygen availability”. By studying EPO gene regulation in transgenic mice and, later, in human cells, Dr. Semenza discovered hypoxia-inducible factor 1 (HIF-1), a protein that controls the production of EPO by cells from the kidney and other organs [68]. This discovery has enormous clinical relevance because HIF regulates the production of endogenous EPO in vivo. Patients treated with a new class of drug called hypoxia-inducible factor prolyl hydroxylase inhibitors (HIF-PHIs), which inhibit the degradation of HIF-α by prolyl hydroxylases, recover from their anemia with EPO produced at physiological levels rather than the pharmacological levels of recombinant EPO [69]. Therefore, the clinical use of these HIF-PHIs may prevent at least some of the possible off-target side effects exerted by pharmacological doses of EPO [70,71].

## 9. Where There Is “Smoke” There Is “Fire”: The Case of Endless Lawsuits

The great economic profits expected by the commercialization of EPO led Amgen to face several lawsuits. The first and most bitter of these lawsuits was that between Amgen/Kirin and the Genetics Institute and its Japanese partner Chugai. 

With the assistance of Dr. Takaji Miyake (who, as mentioned before, was also the first author of the paper that reported the purification to homogeneity of EPO from human urine by the Goldwasser laboratory [11]), the Genetics Institute published the purification and cloning of the human EPO gene a few months before Amgen [61], received 17 December 1984, and accepted 30 January 1985 vs. [31] communicated by Dr. Robert Schimke on 14 May 1985, and published in November 1985. The two studies cloned EPO using an overall similar approach; both of them used degenerated primers designed after amino acid stretches from the protein sequence purified from the urine to fish libraries of human fetal liver cells. Jacobs et al. used these sequences to isolate the EPO cDNA from expression libraries of the human fetal livers, the organ thought at that time to be responsible for EPO production during embryogenesis [72]. Lin et al. [31] instead, cloned the gene from a fetal DNA library provided by Tom Maniatis, expressed this genomic clone in mammalian CHO (Chinese hamster ovary) cells, and then used expression libraries obtained by the transfected CHO cells to isolate the EPO cDNA. 

Both Amgen and the Genetics Institute obtained their first EPO patents in 1987. The Genetics Institute US patent was obtained on 30 June 1987 (U.S. Patent No. 4677195, the “195 patent”). This patent claims both homogeneous EPO characterized and a method for purifying human EPO. An Amgen scientist, Dr. Lin, obtained a US patent on 27 October 1987 (Patent No. 4703008, the “008 patent”). This patent claims purified and isolated DNA sequences encoding erythropoietin and host cells transformed or transfected with a DNA sequence.

There were differences in the knowledge that was used as a source of the patent by the two institutions. Thanks to the fact that the University of Chicago never wanted to patent the sequence of the human EPO protein purified from urine by Gene, the patent from the Genetics Institute covered the sequence and biological activity of the protein present in the urine. Instead, the patent by Amgen covered the sequence of the protein secreted by the CHO line molecularly engineered to express the human gene. Since the process of secretion of EPO from the kidney into the urine involves the proteolytic cleavage of the last three carboxy-terminal amino acids of the protein, the sequences described in the two patents are different only for these three amino acids. The similarity of the purification strategies, the close timing of the two publications, and the great economic interests associated with the clinical use of EPO led to a bitter patent dispute that spanned three-and-a-half years and had controversial outcomes over time. The case was heard by a court in Massachusetts in 1989 that validated the central claims of both Amgen’s and the Genetics Institute’s patents [73]. The two companies appealed this sentence with the federal court in Washington DC, against the expectation that would decide that the patent would be cross-licensed to the two companies, affirmed in 1991 that only Amgen’s patent was valid and enforceable because the Genetics Institute had failed to show that they had purified the EPO which was physiologically active in the body [74,75,76]. This sentence awarded Amgen with 17 years of monopoly to sell EPO in the US. Independent from this legal battle, Amgen received approval for Epogen from the FDA in June 1989 for the treatment of anemia in patients with chronic kidney failure. Along with marketing approval, the FDA also awarded Epogen “orphan drug” status under an Act that provided companies with incentives to develop drugs to treat rare diseases and had approved the use of EPO for the treatment of anemia in HIV-infected patients receiving zidovudine (AZT) (Figure 1). The victory in the legal patent battle, the FDA approval to sell EPO for clinical use, and its recognition as an “orphan drug” status determined that the value of Amgen stocks skyrocketed overnight. 

More recently, Amgen, together with Johnson&Johnson or other partners, fought additional bitter legal battles over EPO patent rights: one against Transkaryotic Therapies (TKT; Cambridge, MA, USA) and Aventis Pharmaceuticals (Bridgewater, NJ, USA) that was won on 19 January 2001 [77,78,79] and another one against F. HOFFMANN-LA ROCHE LTD that was won in 2009 [80]. In general, however, Amgen’s policy has been to prevent potential patent disputes with small Biotech companies that had cloned and expressed human EPO under various conditions by buying their potential patent rights (see, as examples, [62,81,82]). 

In conclusion, since 1991, Amgen has retained the rights toward EPO under the US United States Patent number 4703008 (008), which was extended to cover also the most effective glycosylated form of the protein [83,84,85] for 17 years (until 2008). 

## 10. Additional Clinical and Non-Clinical Uses of Recombinant Erythropoietin

As impressive as it was for the efficacy of recombinant EPO to reduce the transfusion needs of patients with chronic kidney failure, the size of this patient population is not sufficiently great to justify the enormous commercial expansion of the drug. Although the status of the orphan drug extended the FDA approval to the use of EPO to treat anemia induced by AZT in patients with AIDS, the size of this patient population is also relatively small. Also small is the size of the patients requiring autologous blood transfusion to improve the outcomes of elective surgery. In fact, concerns about the safety and availability of the blood donated for transfusion suggested the “brilliant” therapeutic idea of using recombinant EPO to boost blood production in patients scheduled to undergo elective surgeries in order to collect their blood before the surgery and to infuse it as a form of autologous transfusion, if needed, during and immediately after the surgery [86]. The big booster of EPO sales was represented by cancer patients and by the personal use of normal individuals. 

The observation that recombinant EPO improves the exercise capacity of anemic hemodialyzed patients who were still transfusion independent [87] paved the way to devise therapeutic uses for EPO to improve the quality of life of patients with a great variety of pathological conditions, ranging from the anemia of the elderly caused by chronic inflammation to that sustained by the chemotherapy used to halt disease progression in cancer patients. It was the large number of cancer patients, including those with breast cancer, who soon became the biggest EPO users. Both the patients and the physicians loved to use EPO to boost red cell production after chemotherapy; the patients felt “great” and active and were more likely to be compliant with chemotherapy. An excellent review of the therapeutic use of EPO outside the uremic settings may be found in [86]. 

In 1989, Dr. Alan D’Andrea, in the laboratory of Dr. Harvey Lodish and in collaboration with the Genetics Institute, published the cloning of the murine EPO receptor (EPO-R) [87]. Harvey Lodish, in collaboration with the Genetics Institute, published the cloning of the murine EPO receptor (EPO-R) [88]. The cloning of the murine gene was soon followed by that of the corresponding human gene, which was independently published by two laboratories [89,90]. This discovery led to the determination of the number of EPO-R expressed by erythroid progenitor cells [91,92]. It soon became clear that EPO-R is expressed by most hemopoietic cells and that its erythroid specificity is determined by the fact that it is present at detectable levels only on the surface of erythroid cells. In addition, even erythroid cells display very few EPO-R binding sites on their surface (400–800 binding sites per cell). EPO-R is, therefore, capable of delivering its signal with a very low number of EPO-EPO-R occupancy per cell. This high potency is explained by the observation that in addition to being active as a preformed homodimer [26], EPO exerts its biological functions as a heterodimer with other receptors such as cKIT, the receptor for the hematopoietic growth factor stem cell factor [93,94] and the receptor for glucocorticoids [95]. On the one hand, the interaction between EPO-R and cKIT may be physiologically important to guide erythropoiesis in the fetus prior to the production of EPO [96]. On the other hand, the synergism provided by the interaction between EPO-R and the glucocorticoid receptor may explain the erythrocytosis experienced by patients with Cushing’s syndrome who express high levels of glucocorticoids in their plasma [97].

In the hematopoietic system, EPO-R is expressed at low levels by B-cells and dendritic cells [98]. These observations suggest that EPO may exert immunomodulatory functions. EPO-R expression in B-cells has also been suggested to facilitate bone remodeling [99]. Furthermore, EPO-R is also expressed by white adipocytes, and the cooperation between the downstream signaling activated by EPO-R in B-cells and in adipocytes may contribute to maintaining bone homeostasis within the bone marrow microenvironment [100]. 

EPO-R also forms heterodimers with the beta chain of the interleukin-3 (IL-3) receptor [101,102]. Although it is unclear whether this heterodimer is active in erythroid cells, it is well-established that it plays an important role in the regulation of tissue regeneration. The EPO-R/beta-chain IL-3 heterodimer is optimally activated by a small EPO peptide which, although not active in erythroid cells, has been investigated by Dr. Antony Cerami’s laboratory for use in tissue regeneration, including repair from ischemia, and neuroprotection [103]. The fact that EPO-R is expressed in the brain and that is required for proper development [104] and the additional non-erythroid effects of EPO discovered by several investigators raise concerns about whether the improvement in the quality of life experienced by anemic patients treated with recombinant EPO is directly due to the greater tissue oxygenation provided by the normalized hematocrit or to off-target effects of the drug [105,106]. Among all the reported non-hematopoietic activities of EPO, the only one that moved to clinical practice and standard of care is the treatment of hypoxic–ischemia encephalopathy for the newborn [107] (Figure 1).

The misuse of EPO in sports (EPO doping) is widely known. It is a dark story fueled by the combined interests of athletes (who aimed to reach fame), National States (to boost national pride), and pharmaceutical companies (to increase profits). The personal use of EPO by healthy individuals is widely practiced in professional, semi-professional, and even amateur sports. The story in amateur sports is less discussed in public arenas and involves the occasional use of EPO by individuals who just want to show off in city marathons or in soccer games organized by local parishes, thanks to the complicity of the managers of private health clubs. The fact that this practice raises limited ethical concerns is exemplified by a phone conversation I overheard during my residency in Seattle, which occurred between John and his academic boss, Dr. Phil Fialkow. It was 1986 when John’s lab was flooded with Epogen to be used for the clinical trial in patients with chronic renal failure. Dr. Fialkow, a great geneticist and member of the Institute of Medicine, now the National Academy of Medicine, who had discovered the clonal origin of Cancer [108], was the Head of the Department of Medicine, which included the Division of Hematology chaired by John. As described in his obituary in the University of Washington magazine [109], Dr. Fialkow had a passion for site trekking in Nepal and every year organized with his wife a trip to a different National Park in that region of the Himalayas, all of which were at high altitude. The site trekking lasted longer than expected because it included stopovers at increasing altitudes to allow the body to raise its hematocrit levels to cope with hypoxia. That year, Dr. Fialkow had a busy schedule and wanted to make the trip as short as possible. Being the brilliant clinical geneticist he was, he was quick to conceive that he could reach his goal by using EPO instead of exposure at increasing altitude to boost his red cell production. At that time, EPO was not for sale, and Dr. Fialkow called John to obtain some of the Epogen provided by Amgen for the patients. The call was transferred to the fellow’s office where my husband and I were discussing with John the experimental plan for the week. We could not hear Fialkow’s words, but we clearly heard John’s response: it is not allowed. Although that was the end of Fialkow’s proposition, “chats” with my medical students assured me that each of them knew somebody who knew somebody who had occasionally taken EPO under the supervision of a personal trainer. 

The story of “doping” in professional and semi-professional sports is widely documented in the literature. It started as soon as the Adamson lab, among others, had published that the plasma of animals exposed to hypoxia increases the hematocrit when injected into anemic animals [44]. From that moment, athletes around the world, under medical supervision, started to experiment with exposure to hypoxia or transfusion with plasma from subjects exposed to hypoxia as a form of doping. The practice started with horse raising, but it was expected that soon, also men would start using EPO to improve their “physical performance”. I remember a conversation that occurred in 1987 near the coffee machine, a popular conversation place in the Adamson’s lab, between him and Dr. Joan Egrie, during one of her regular visits to review the data of the trial. They discussed adding “a tracer” in the Epogen vials to prevent its misuse. This idea did not go anywhere mainly because Amgen feared that any unnecessary addition to the Epogen formulation may increase the risk of side effects. 

It was, however, the paper published by Eschbach and Adamson in 1990 demonstrating that recombinant EPO improves the exercise capacity of anemic hemodialysis patients [87] that misguidedly encouraged the use of EPO doping in sports. This misuse was practiced across countries. In Italy, it was implemented by a physician, Dr. Francesco Conconi, the Hematologist at Ferrara University, who had made an international name for himself for his studies on the genetics of beta-thalassemia in the population of Ferrara [110] Dr. Conconi had developed a test, known as the Conconi Test, that on the basis of the correlation between physical exercise and heart-beat calculated the hypoxia level of the body and which was widely used to measure the performance of athletes [111]. In 1980, he offered his medical assistance, and that of his university staff, to the Italian National Olympic Committee (CONI), to improve the performance of the Italian athletes of several disciplines, ranging from cycling, rowing, and swimming. The proposal was accepted and included a strong program of research in sports medicine supported by public fundings. During the period of his medical assistance, the national Italian athletes excelled in numerous international competitions including the 22 honor medals collected at the XVII Winter Olympic Games when the Italian cross-country skiing athletes who won the nine medals had hematocrit levels greater than 50%, raising a strong suspicion that they had been treated with EPO. In 1992–1994, when these facts occurred, the detection of such high EPO levels did not have any consequence and did not mandate, as it is today, the suspension of the athlete from competition [112]. It is interesting that sports regulatory agencies have implemented the suspension of athletes found guilty of doping on the basis of safety rather than moral considerations. As officialized by the sentence n. 533–2003 of the Ferrara tribunal, the medical assistance provided by Dr. Conconi to the Italian National athletes was basically doping with recombinant EPO [113]. The tribunal calculated that over the years at least 450 Italian athletes were doped by Dr. Conconi with EPO and some of them, such as the cyclist Pantani, died of cardiovascular complications relates to the use of this hormone [112]. It is somewhat ironic that Prof. Conconi was also among the official medical experts for doping of the Italian Judiciary System. During my fellowship in Seattle, I was involved in a project coordinated by Dr. Ginny Broudy which attempted to characterize EPO-R in human erythroblasts expanded in culture. The project was based on the large amount of Epogen available in the lab which allowed, after chemical labelling with ^125^-Iodine, to be used as a probe to identify its binding protein (i.e., EPO-R) by Western blot. As later discovered by Dr. Broudy [91], the amount of EPO-R expressed by erythroid cells is too low for this brilliantly conceived experiment to succeed. However, one of the lanes of the blot contained ^125^-I EPO as control and I recall noticing that EPO migrated as a single band, against the smear expected for the natural protein present in the serum which is not, as the recombinant product, homogeneously glycosylated. When, in 1987, I returned to Italy at the Istituto Superiore di Sanità, which is the referral Center for the Italian Ministry of Health for wellbeing practice, I wrote an email to my superiors stating that, in my opinion, a Western blot of an athlete’s plasma would allow to discriminate whether the serum contained the natural or the recombinant protein. My email was ignored (for my own good since I had time to develop more interesting science), probably because of a paper which had not found differences between the physicochemical properties of “pure” recombinant and natural EPO [114]. This paper, however, overlooked that the purification process, which, by definition, selects for homogeneously glycosylated proteins, was a confounding factor when evaluating the property of the protein in the circulation. We had to wait until 2003 for the demonstration that two-dimensional gel electrophoresis clearly discriminates the “recombinant” vs the “natural” form of EPO [115]. A review of the progress made to detect EPO doping in sports may be found at [116].

A twist in the story of EPO-doping was provided by the seven-time Olympic medalist (three of them gold) of cross-country skiing, Eero Mäntyranta. The hematocrit levels of this Finnish athlete were extremely high and the fact that his EPO levels were low did not completely tame the suspicion of doping because physicians have ways, diuretics and/or saunas, to reduce the EPO levels in the blood before testing [117]. The name of Eero was cleared up when Dr. Albert de la Chapelle discovered that he, and his family, harbored a genetically mutated form of EPO-R with increased sensitivity [118,119]. It was later discovered that gain-of-function mutations of EPO-R are not rare (see, as an example, [120]. In addition, genetic mutations which induce erythrocytosis by increasing EPO production have also been discovered. One such mutation is the Chuvash mutation, named after the geographic region where it was discovered for the first time. This is the point mutation 598C>T in the gene encoding the von Hippel–Lindau (VHL) tumor-suppressor protein, one of the elements of the kidney hypoxia response that regulates EPO production [121]. Individuals with 598C>T VHL experience constitutive activation of the hypoxia pathway and produce high levels of EPO [122]. This mutation is the most common mutation associated with familiar erythrocytosis and is distributed worldwide spread from a single founder 1000 to 62,000 years ago [123]. The founder “effect” is probably due to a positive selection because the mutation at the heterozygous state protects from anemia. It is found at an incredibly high frequency in the small island of Ischia in Italy, where my husband was born, and is particularly frequent in his family which represent one of the largest pedigrees included in the paper [124]. My father-in-law was a heterozygous Chuvash and his phenotype summarizes all the advantages expected to be provided by high EPO levels (Figure 4); he maintained hematocrit levels above normal (54%) until he died at an old age (99), had healthy bones (we had visual proof of how robust his bones were when we attended the procedures to exhume his body 5 years after his death as customary for the island), strong muscles, and was extremely resilient to fatigue. The joke in the family is that he dug down the last 8 mt holes necessary to plant new grapes and acted as the “human” jack to raise his Ape Piaggio to change the posterior tires until the age of 88. However, in the homozygous state, the mutation is associated with increased morbidity and risk of thrombotic events [125] and the epidemiology of this mutation provided the strongest indications on the negative side effects of high EPO levels sustained by doping which, as experienced by the Chuvash, include enhanced likelihood of stroke, myocardial infarction, thrombosis, and an increase in total peripheral vascular resistance [126].

The competitive advantage provided by genetic mutations, which increase hematocrit levels, raised the question of whether sports committees around the world should implement distinctive categories similar to those established based on gender and based on genetic makeup [129]. Since each individual is a “unique” genome, there is no finite end to the establishment of “individual categories” and the decision was made not to take any further action. 

In 2002, the British pharmaceutical company Oxford BioMedica developed Repoxygen, a viral vector harboring a modified human *EPO* gene under the control of genes encoding proteins of oxygen homeostasis (e.g., HIF-1, HIF-2), as a potential drug for the treatment of anemia associated with the chemotherapy used in kidney cancer [130]. The advantage of this drug over protein-based products is that once administered intramuscularly, the virus integrates into the genome of the host cells and produces EPO as long as the body is exposed to low levels of oxygen. EPO production is then turned off when the oxygen concentration returns to normal. Since this retrovirus simulates the Chuvash mutation, it is very difficult to distinguish whether high EPO levels in an individual are due to a congenital or a retrovirus-induced mutation. In 2006, Repoxygen attracted the attention of the sports world when it was found to have been administered to young female runners in Germany in order to maintain constant expression of EPO in their muscle cells. Repoxygen is prohibited under the World Anti-Doping Code 2009 Prohibited List [131]. This retrovirus initiated the era of “genetic doping”, which is now used to increase the levels of many additional “performance-potentiating” hormones. 

## 11. The Clinical Use of EPO May Not Be So Benign after All: Development of Neutralizing Antibody and Acceleration of Disease Progression in Patients Treated with EPO

### 11.1. The Case of EPO Antibodies

The observation that the replacement of insulin from animal sources with the recombinant product had greatly decreased but did not eliminate the incidence of the development of antibodies against insulin in diabetic patients, making them resistant to the therapy [132] raised concern that also patients treated with recombinant EPO may develop antibodies against the protein and become transfusion-dependent for the rest of their life. However, by contrast with diabetic patients who may develop alloantibodies against insulin prior to exposure to the recombinant protein [133], the occurrence of anemia due to the production of alloantibodies against EPO has never been reported. In addition, the natural and recombinant EPO are almost identical [57,114], greatly reducing the probability that the body would generate antibodies.

For many years, hundreds of thousands of patients were safely treated with recombinant EPO with no sign of having developed antibodies. However, in 1999, when I was working at the New York Blood Center, I was approached by Dr. Jim Bussell of Cornell University, who was interested to know whether the serum of 12 AIDS patients who had been treated with recombinant EPO and had returned to be transfusion-dependent contained antibodies against EPO. Since the possible presence of the AIDS virus made the test tricky, I decided to perform it myself. The test involved the addition of recombinant EPO to the patient’s serum, incubation of the serum with beads that bind immunocomplexes, and test of the supernatant for the presence of EPO by biological assays. If the serum contained antibodies against EPO, these antibodies would have bound EPO, and the protein complex would have been removed by the beads, making the serum unable to sustain the growth of murine erythroid colonies in culture. Not surprisingly, after incubation with the magnetic beads, the serum of all 12 patients became unable to stimulate colony growth (AR Migliaccio, unpublished results). I sent this information to Dr. Bussel, and for me, the story stopped there. The story gained momentum in Europe, where a cohort of patients treated with recombinant EPO developed pure red cell aplasia and became transfusion-dependent because they had developed antibodies. The patients were monitored for the development of anti-EPO antibodies thanks to the excellent work of Dr. Nicolle Casadevall, which was first published in 2003–2004 [134,135].

Overall, the number of patients who developed antibodies against EPO was quite small (<150 patients) when compared with the large numbers of those being treated with EPO, which were mostly Europeans. In addition, most of them had received Eprex/Erypo or other formulation of Epoetin alpha, which, by contrast with the US formulation and in accordance with new safety requirements of EMEA, did not contain human serum albumin. It has been proposed that the human serum albumin-free formulation is less stable, allowing aggregates of EPO molecules to form, aggregates which had been already demonstrated to be the main cause for the development of antibodies against other therapeutic recombinant proteins [136]. While the implementation of packaging guidelines to prevent the formation of protein aggregates has solved the problem of antibodies for therapeutic proteins, this story has a moral: in the development of pharmaceutical recombinant products, science is great, but at the end of the day, small production details are of paramount importance for the safety of the product. Regulatory agencies worldwide spend a great part of the time necessary to authorize a new pharmaceutical product for clinical use in the analyses of the production process. 

### 11.2. Acceleration of Disease Progression in Cancer Patients Treated with EPO

In his book “A Bloody Long Journey”, Gene Goldwasser had already devoted a short chapter to the “Misuse of EPO”, concerning both competitive sports and in the clinic. He states, “some of the reports I’ve seen suggest that administration of EPO to anemic patients was biased by the wish to sell more of the hormone than was needed clinically. That would not surprise me”. The EPO misuse for doping in sports was discussed earlier. Here, we will address the clinical misuse of the hormone. 

The clinical misuse of EPO was fueled by the enthusiasm of the patients, who felt “great” with higher hematocrit levels, by the physicians, who saw great improvements in the quality of life of their patients, and by the greediness of the pharmaceutical companies. Greater pharmacological doses of EPO, however, were not without off-target effects. Already in hemodialyzed patients, several investigators noticed that, as described for patients homozygous for the Chuvash mutation [125], pharmacological doses of EPO increase the risk for hypertension and stroke [137,138]. This observation was late to be implemented into the restriction of the use of EPO in clinical practice. It was the misuse of EPO in cancer patients that raised the strongest and more controversial concerns. The debate was: do cancer cells express EPO-R and proliferate at pharmacological doses of EPO? In an email written to me on 8 January 2024, Dr. Noguchi stated: “David Hankins (the senior author of her paper on the cloning of human EPO-R) [90] was not happy to learn that the EPO-R promoter was active in Hela cells”. Hela is an epithelial cell line derived from the uterus cervical cancer of a patient by the name of Henrietta Lacks [28,137]. The fact that the sequences that lead the expression of the EPO-R gene are active in epithelial cells indicates that they are not lineage-specific. The generation of antibodies against EPO-R did not answer the question. These antibodies were first generated by Dr. Alan D’Andrea [139], and two of them (M-20 and C-20) became commercially available and were widely used by the scientific community to detect EPO-R (Figure 5).

A study conducted by Dr. Steve Elliott at Amgen, however, challenged the specificity for EPO-R of these two antibodies [140], raising concerns that the studies that had detected the receptor in non-erythroid cells may not be valid. We had the opportunity in our lab to test the M-20 and C-20 antibodies against protein extracts of human erythroblasts expanded in vitro using extracts for the BaF3 cell lines overexpressing human EPO-R as control (Figure 5 and [95]). Indeed, the two antibodies detected multiple bands from the BaF3 line overexpressing human EPO-R. The C-20 antibody was clearly superior to M-20 for the lower number of non-specific proteins recognized and for the intensity with which it recognized the EPO-R specific band in BaF3/EPO-R cells (>100-fold higher). However, most of the bands detected by C-20 were not expressed by the naïve cells, raising concerns about whether at least some of them may represent EPO-R degradation products. C-20 also consistently detected two bands of the expected molecular weight for EPO-R [95] that co-migrated with the EPO-R specific band of the BaF3/EPO-R cells in primary erythroblasts obtained from three normal donors. The levels of EPO-R expressed by primary erythroblasts were, however, very low (<100-fold than in BaF3/EPO-R), suggesting that it may be challenging with this antibody to detect EPO-R in non-erythroid cells that express the protein at even lower levels. The band of the molecular weight corresponding to EPO-R detected by both C-20 and M-20 was not expressed by naïve BaF3 cells, confirming that it was EPO-R. In this regard, the 56KD band identified by M-20 was recently confirmed to represent the EPO-R protein by mass spectrometry by Lamon et al. [141]. For its high sensitivity, the antibody C-20 was used to detect EPO-R until recently (see, as an example, [142]).

Several investigators, mostly from Europe, reported biochemical (activation of signaling pathway) and biological (growth) of non-hematopoietic tumor cells in vitro [143,144,145,146,147]. Few of them, however, failed to detect the activity of EPO on cancer cells [147,148]. The controversy over the biological activity of EPO in cancer cells and the supposed non-specificity of the M-20 and C-20 antibodies led to a long debate about whether cancer cells express the receptor for EPO. To quote Dr. Terry Lappin’s “words” (e-mail to ARM in February 2024): “There has been a long debate and I think it is fair to say that generally the Pharma companies set out in rather cavalier fashion to deny, or at least downplay, the existence of EPO receptors in tumor tissue, whereas many investigators were much more cautious”. A flavor of the ongoing debate on EPO and cancer may be found at [149,150]. Based on the paper by Dr. Elliott and on the testimony of Dr. Lodish, who was also the legal advocate for Amgen in patent disputes [76,77,78], the FDA did not take any decision on an eventual restriction of the use of EPO in cancer patients for a long time. Lodish, who was also the legal advocate for Amgen in patent disputes [77,78,79], said the FDA did not make any decision on an eventual restriction of the use of EPO in cancer patients for a long time. As always, when biology fails to provide answers, epidemiology takes the lead. A meta-analysis by Dr. Julia Bohlius and colleagues of the follow-up of 13,933 cancer patients included in 53 trials finally established that treatment with erythropoiesis-stimulating agents in patients with cancer increases mortality and worsens overall survival [150,151]. This meta-analysis was associated with the results of two clinical trials that evaluated the risk of cardiovascular diseases and benefits for the quality of life of patients with chronic kidney failure receiving EPO, the CHOIR and CREATE study [152,153]. CHOIR (ClinicalTrials.gov number, NCT00211120) was an open-label trial that recruited 1432 patients with chronic kidney disease who were being treated with Epoetin alpha. The patients were randomly divided into two groups, which received doses of Epoetin alpha, which raised hemoglobin levels to either 13.5 g/dL or 11.3 g/dL. The patients were studied for 16 months, and the endpoints of the study were death, myocardial infarction, hospitalization for congenital heart failure, and stroke. This study convincingly demonstrated that higher EPO doses increase the risk of vascular complications without any benefit for the improvement of the quality of life. The design of the CREATE study ClinicalTrials.gov number, NCT00321919, was similar to that of CHOIR and, although it included a smaller number of patients (only 603), had a longer follow-up (3 years). CREATE independently reached conclusions very similar to those of CHOIR. Based on this overwhelming clinical issue, finally, in 2011, i.e., two years after the expiration of Amgen’s patent on recombinant EPO, the FDA decided to implement major changes in the safety of the use of EPO drugs. For its implications for cancer patients, this FDA decision raised great attention from the media [154] and represented a major shift in the policy of public funding agencies, which began to support investigators to study EPO-R. In Europe, an EU FP7 Health Grant was awarded to the EpoCan Consortium of collaborators from Austria, Germany, Israel, the Netherlands, Spain, Switzerland, and the United Kingdom to support the study of long-term risks and advancement towards better Epoetin-driven treatment in 2011. Results supported by this award were published in 2015 [155].

By contrast with other experts, Dr. Adamson had always been very vocal about the possible side effects of the excessive use of EPO by patients [137]. Maybe this was the reason why Dr. Binder, the former CEO of Amgen who guided the process of EPO commercialization, was even unable to correctly spell the names of the scientists who contributed so much to the financial success of the company. In fact, in the first paragraph on page 94 of his book, Dr. Binder states that: “The Epogen’s phase I-phase II clinical trial commenced at Seattle’s University of Washington Medical Center in December 1985 under the direction of Dr. David Dale, a prolific investigator in the area of anemia [33]”. By contrast, as mentioned earlier, the first Epogen clinical trial was performed by Drs. Eschbach and Adamson, and it was Dr. Adamson who had extensive previous preclinical experience of anemia and EPO at the University of Washington. Dr. David Dale, a Professor of Medicine at the same University, is actually an expert in chronic neutropenia and performed the first clinical trial with recombinant G-CSF for Amgen in the genetic disease cyclic neutropenia [156]. I know this trial very well because I was responsible for determining the hematopoietic progenitor cell content of the bone marrow and blood of the patients during the treatment [157]. My study demonstrated that treatment with pharmacological doses of G-CSF does not lead to hematopoietic stem cell exhaustion, a serious concern for growth factors, such as G-CSF, which are active on these cells and may lead to bone marrow aplasia. Of note, my data were also the first to show that the number of progenitor cells in the blood increased soon after each G-CSF treatment, indicating that this drug induces hematopoietic stem cell mobilization. Short-term G-CSF treatment was later exploited to induce the mobilization necessary to collect from the blood hematopoietic stem cells in numbers sufficient for engraftment after transplantation [158,159]. This practice has greatly facilitated the use of autologous stem cell transplantation to protect the ability of patients with solid cancers to produce blood while undergoing chemotherapy [160]. 

## 12. Overall Summary (See Also Figure 1)

The journey in search of the pharmacological cure for anemia that started in 1909 reached its acme in 1995–1987 with the cloning of the human EPO gene, the first clinical trial that proved the efficacy of recombinant EPO to correct the anemia of patients with chronic kidney failure and the approval of the clinical use of EPO by the FDA. This acme included a close collaboration between Biotech companies, basic scientists, and clinicians. In fact, during these years, Dr. Lin, in addition to cloning human EPO, cloned and helped to produce high levels of recombinant monkey EPO [51], a product with no commercial value per se but that was instrumental for performing pre-clinical studies in the baboons and rhesus monkeys to identify pharmacological treatments for Sickle Cell Disease and Thalassemia [60,161,162]. This acme was followed by a plateau that lasted from 1988 to 2006, during which the pharmaceutical industry took almost complete control of the clinical use of EPO and expanded this use to the treatments of other hematological and non-hematological conditions. However, the great number of patients being treated with EPO enabled the assessment of its negative side effects (increased risks of cardiovascular diseases and acceleration of cancer progression). This awareness led in 2011 to new FDA safety regulations on EPO use. The discoveries by Dr. Semenza, Ratcliffe, and Kaelin, recognized with the Nobel Prize for Physiology or Medicine in 2019, led to the development of a new category of erythroid stimulating agents. These drugs are small-molecule hypoxia-inducible factor prolyl hydroxylase (HIF) inhibitors, which, in contrast with EPO, which is injected, can be formulated into orally active pills and can be altered to titrate the required erythropoietic response. They simulate reduced tissue oxygen pressure, thus stimulating the production of physiologic levels of endogenous EPO by the kidneys and liver. Clinical trials with these compounds demonstrated that they are at least as effective as recombinant EPO in treating or correcting anemia in non-dialysis and dialysis patients. These trials, however, did not demonstrate superiority in safety outcomes, and in some trials, as originally suggested by Dr. Semenza, these outcomes were even worse [163]. Therefore, the exciting journey of erythroid stimulating agents to treat anemia is not over yet, and new discoveries are expected to be found behind Alice’s looking glass. 

## Figures and Tables

**Figure 1 biomolecules-14-00408-f001:**
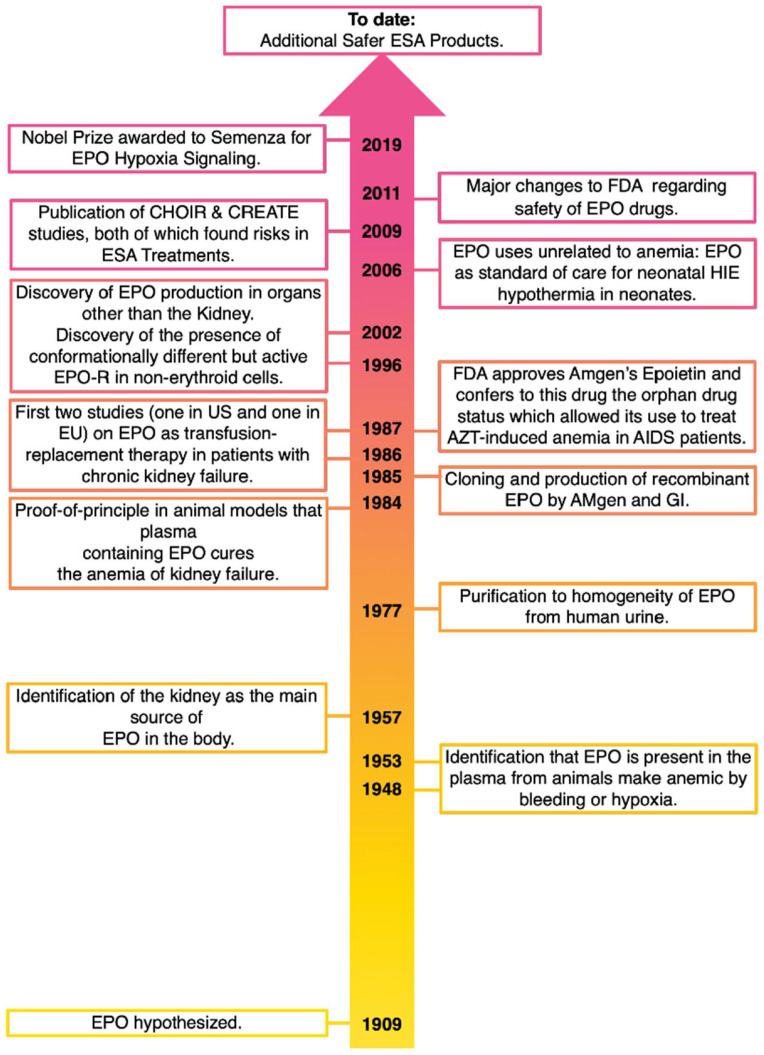
Timeline of the milestones of the Erythropoietin field from its hypothesis in 1909 to the first clinical trial in 1986 to the discovery of new erythropoiesis-stimulating agents (ESA, such as inhibitors of hypoxia signaling) in 2019.

**Figure 2 biomolecules-14-00408-f002:**
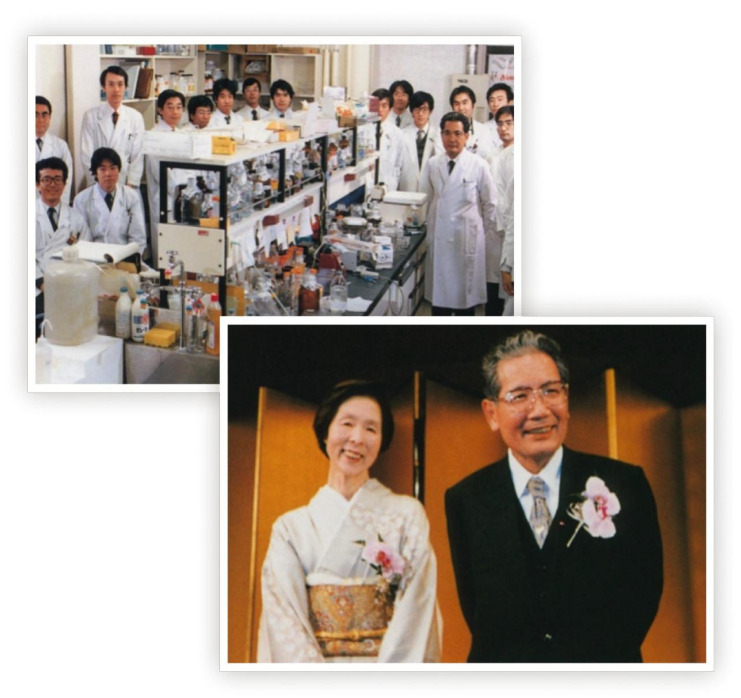
Dr. Fumimaro Takaku and the Japanese School of Hematology who inspired the research on EPO in Japan. The photos may be found online at [45].

**Figure 3 biomolecules-14-00408-f003:**
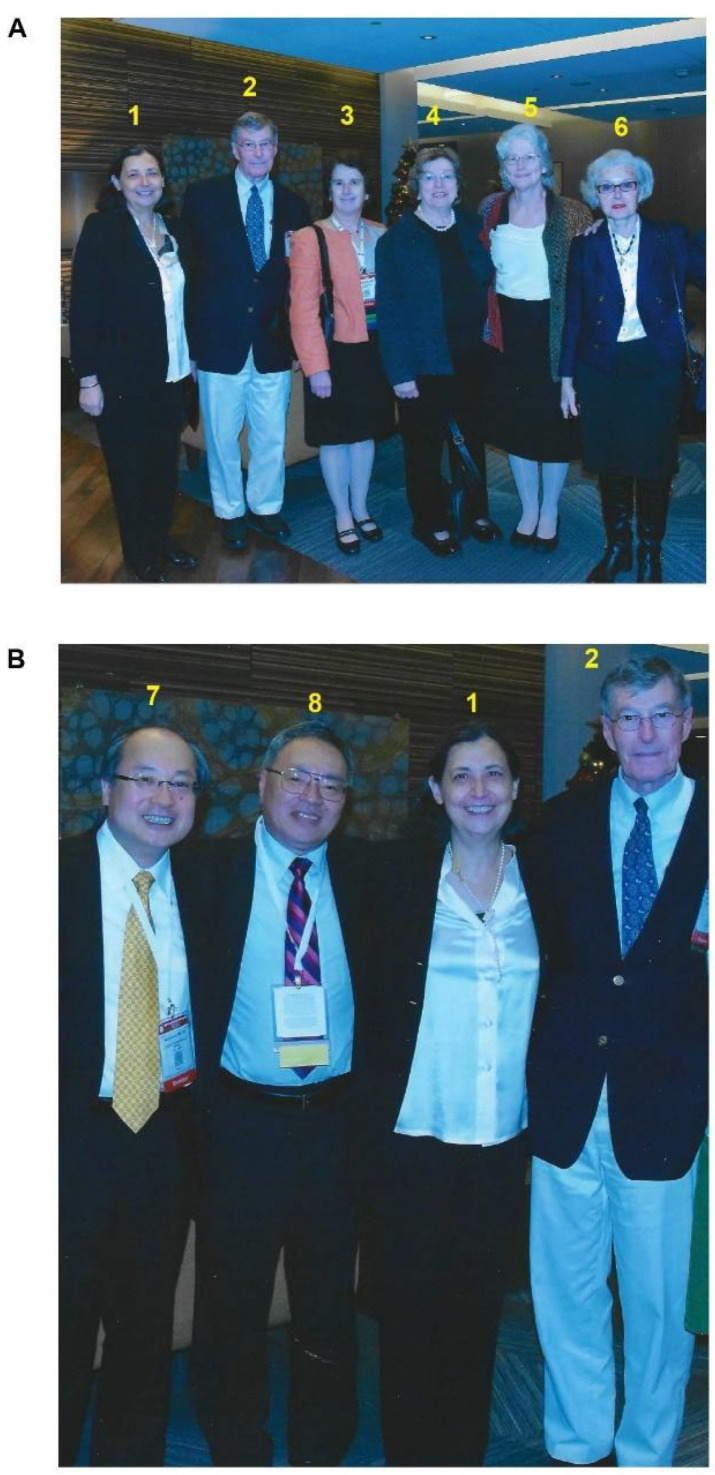
The impressive team of the Division of Hematology, directed by Dr. Adamson, which made an enormous contribution to the discovery of erythropoietin and its clinical application. (**A**) The Erythropoietin Team of the division of Hematology. (1) Dr. Migliaccio, the author of this review who has collaborated with Dr. Adamson for more than 10 years; (2) Dr. Adamson, the mind behind the clinical use of erythropoietin to treat the anemia of patients with chronic kidney failure; (3) Dr. Janis Abkowitz, who studied the effects of EPO in an animal model of leukemia [56] and succeeded Dr. Adamson as Division Chief; (4) Dr. Rebecca Haley, the clinical fellow who cared for the patients recruited in the first clinical trials with recombinant EPO; (5) Dr. Ginny Browdy, who as clinical fellow purified the human EPO gene [57] and characterized the expression of its receptor on human erythroid progenitor cells [58], and (6) Dr. Thalia Papayannopoulou [59,60]. (**B**) The Japanese connection. Dr. Adamson (2) with (8) Dr. Akihiro Shimasaka, the Scientific Director of Kirin Pharmaceutics, and (7) Dr. Norio Komatsu, presently a retired Professor of Hematology at Tokyo University, who is just one of the numerous fellows who were supported by Kirin to learn erythropoiesis in the Adamson laboratory. The photos were taken by Dr. Akihiro (Sam) Shimasaka at the dinner organized at ASH 2015 by the past members of the Division to celebrate the mentorship award conferred by the society to Dr. Adamson one year earlier.

**Figure 4 biomolecules-14-00408-f004:**
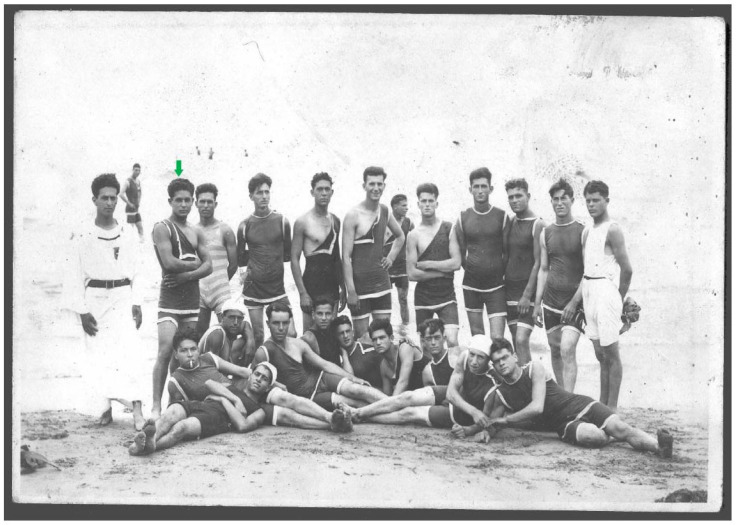
Photograph of an individual heterozygous for the Chuvash mutation highlighting his robust corporature that makes them resilient to fatigue. The arrow points to the father-in-law of the author, who, after having joined the Italian Navy at the age of 16, was sent in 1920 to Tientsin, a Nord-eastern region of China, at that time an Italian colony, as part of a contingent composed of a hand-full of sailors (the other sailors in the photo) in charge of protecting the railroad system against the turmoil of the Chinese revolution [127,128].

**Figure 5 biomolecules-14-00408-f005:**
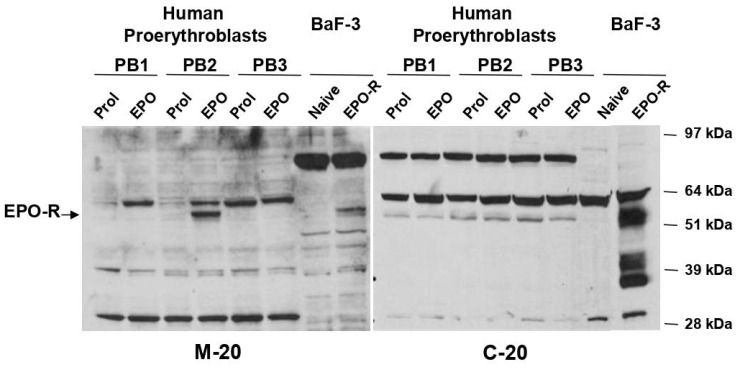
The controversial poor specificity of the antibodies against the human erythropoietin receptor (EPO-R) available in 1990–2000. Western blot analyses with the antibodies commercially available in 1990–2000, M-20 and C-20 against human EPO-R (catalog number sc-695, and sc-697, Santa Cruz Biotechnology Inc., Dallas, TX, USA) of proteins (30 μg/lane) from human erythroblasts expanded in vitro from three different blood donors (PB1, PB2, and PB3) and from BaF3 cells either untreated (naïve) or transfected with the human EPO-receptor cDNA provided by Dr. Alan [89]. The ligand–antibody signal was revealed with appropriate horseradish peroxidase coupled secondary antibodies (Calbiochem, San Diego, CA, USA). The position of the molecular weight markers (in kDa) is presented on the right, while the position of the band with the molecular weight expected for EPO-R is indicated on the left (ARM, unpublished observations, see also [95]).

## Data Availability

All the data discussed in this paper paper are either disclosed in original publications or available in publicly available data sets.

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
