# Peer review of "Erythropoietin: A Personal Alice in Wonderland Trip in the Shadow of the Giants"

_biomolecules, 2024, doi:10.3390/biom14040408_

Round 1
Reviewer 1 Report
Comments and Suggestions for Authors
Throughout the paper investigators were inconsistently referred to, some had first name dropped, some had first name included (this reviewer’s preference), some referred to as doctor and others with PhDs or MDs not termed Doctor…the author should be consistent throughout the paper
Also:
Line 27 and 124: perhaps “Herculean task” instead of “gigantic job”
Line 61: “in the early 17th century”, or “in 1600” (without century)
Line 65: “for some time” instead of “for long time”
Line 80: might want to mention why the kidney is the “best” organ to regulate red cell production (because it extracts the least oxygen with nearly the greatest blood flow, so that the EPO-producing organ itself isn’t subject to metabolic demands and skew erythropoiesis)
Line 103 and several other places: do you mean dark side(s) or dark site(s)…I suspect the former
Line 115: replace “disorder” with “neoplasm”
Line 120: drop “up”
Line 122: replace “does not return” with “returns”
Line 127: replace “device” with “development”
Line 254: the author might want to also cite the book The Immortal Life of Henrietta Laks, by Rebecca Skoot, in which an even more egregious failure to recognize patients’ rights was undertaken, famous because the outcome of the work at Johns Hopkins University lead to the development of the HeLa cell line, used by countless investigators worldwide since its generation in the early 1950s.
Line 300: you might want to also give credit to Lee Hood, then at Caltech I believe (and who eventually came to UW), who took Genes purified EPO and (protein) sequenced it, allowing AMGEN to design genetic probes for EPO
Line 320: might also mention JWA as chief of hematology at UW
Line 327: not sure why this section is italicized, as well as line 360 onward
Line 353: I think its “exciting”, not “exiting”, and I think its Rebecca Haley, not Halley
Line 420: replace “resident medical fellow” with “hematology fellow”
Line 421: please check you have the name right – I thought Aranesp was the second generation EPO created to allow re-patenting EPO by adding carbohydrate side chains to increase half life and thereby reduce dose frequency….
Line 448 and 619: its University of Chicago, not Chicago University
Line 719, 751: “professional and amateur athletes” instead of “sportsman”
Line 721: “by” instead of “from”
Line 729, 738: Phil Fialkow was not a hematologist, he was a geneticist, and instead of stating “Medicine National Academy of Sciences”, would state “Institute of Medicine, now National Academy of Medicine…”
Line 763: I would absolutely not use the word ‘officializing”, rather, might say “unfortunately, misguidedly, encouraged…”
Line 781: found, not fund
Line 816: might want to also cite an editorial that accompanied de la Chapelle’s paper by Greg Longmore (Nat Genet 1993 Jun;4(2):108-10) because the title of the article is so cool: “Epo Receptor Mutations and Olympic Glory”
Comments on the Quality of English Languageminor changes needed
Author Response
See the submitted file

Reviewer 2 Report
Comments and Suggestions for Authors
This review by Anna Rita Migliaccio is delightful history of the discovery of erythropoietin and its development into a major therapeutic. It tells the story good and bad about the personalities of the researchers who were directly and indirectly involved in the identification, cloning and later clinical trials for Epo. I enjoyed reading it. However it is too long. My suggestions would be to limit the discussion of blood doping and the sections on the erythropoietin receptor.
Author Response
See the submitted file

Reviewer 3 Report
Comments and Suggestions for Authors
This paper is a comprehensive review about the history of erythropoietin from discovery, purification, industrial production and its use in medicine written by an expert who has been in this field for a long time.
As somebody with a long life interest in protein chemistry who has worked in the field of the isolation of peptides and proteins (in the laboratories of Nobel Laureate Dr.Roger Guillemin who passed away one week ago at the age of 100) and Dr.Eric Shooter (epidermal and nerve growth factor) I have much enjoyed to learn many details of the erythropoietin history who lasted over several decades.
The author also talks about about a possible Nobel prize for the tedious work on erythropoietin and comparing it to the prices Dres. Stanley Cohen and Dr.Rita Montalcini received during that era.
I think their price was awarded more for the identification of EGF and NGF and less for the purification which was a relatively easy task since the submaxillary glands contain tons of these proteins compared to neuropeptides in the hypothalamus or erythropoietin in biological fluids.
In the long run the discovery of EGF had also major implications for the progress in oncology (for instance, HER2NEU in breast cancer or EGF-receptor mutations for personalized therapy in lung cancer).
Moreover, one could also argue that Dr.Sidney Pestka should also have received a price for his purification of interferon.
Nowadays, in the clinic we are happy to have recombinant erythropoietin available. Moreover, identification of its receptor and the mechanisms of oxygen sensing has allowed a better understanding of polycythemia and hereditary erythrocytoses.
All this is eloquently described in the submitted review which will be a landmark article for those readers who are interested in the history of medicine.
Author Response
See the submitted file

Round 2
Reviewer 2 Report
Comments and Suggestions for Authors
The authors have revised and improved the manuscript. They have addressed my concerns.